# Effects of Hexagonal Boron Nitride Insulating Layers on the Driving Performance of Ionic Electroactive Polymer Actuators for Light-Weight Artificial Muscles

**DOI:** 10.3390/ijms23094981

**Published:** 2022-04-29

**Authors:** Minjeong Park, Youngjae Chun, Seonpil Kim, Keun Yong Sohn, Minhyon Jeon

**Affiliations:** 1Department of Nanoscience and Engineering, Center for Nano Manufacturing, Inje University, Gimhae 50834, Korea; mjpark9121@gmail.com; 2Department of Industrial Engineering, Bioengineering, University of Pittsburgh, Pittsburgh, PA 15261, USA; yjchun@pitt.edu; 3Department of Military Information Science, Gyeongju University, Gyeongju 38065, Korea; seonpil@gu.ac.kr

**Keywords:** insulators, interface structure, electrochemistry, mechanical properties, ionic electroactivepolymers, acuators, artificial muscle, capacitance

## Abstract

To improve the energy efficiency and driving performance of ionic electroactive polymer actuators, we propose inserting insulating layers of 170 nm hexagonal boron nitride (h-BN) particles between the ionic polymer membrane and electrodes. In experiments, actuators exhibited better capacitance (4.020 × 10^−1^ F), displacement (6.01 mm), and curvature (35.59 m^−1^) with such layers than without them. The excellent insulating properties and uniform morphology of the layers reduced the interfacial resistance, and the ion conductivity (0.071 S m^−1^) within the ionic polymer improved significantly. Durability was enhanced because the h-BN layer is chemically and thermally stable and efficiently blocks heat diffusion and ion hydrate evaporation during operation. The results demonstrate a close relationship between the capacitance and driving performance of actuators. A gripper prepared from the proposed ionic electroactive polymer actuator can stably hold an object even under strong external vibration and fast or slow movement.

## 1. Introduction

Ionic electroactive polymer (IEAP) actuators have garnered significant attention for use in applications such as intelligent robots, biomedical devices, and micro-electro-mechanical systems [1,2,3]. They are lightweight, operate at low voltages, and are suitable for a wide range of environments [4,5,6,7,8]. IEAP consists of two electrodes and a polymer membrane. Various types of electrodes [9,10,11] and polymer materials [12,13,14,15,16,17] have been considered for improving IEAP actuator performance, but further research is required to increase energy efficiency, reliability, and durability. Recently, the interface between the electrode and polymer membrane has attracted research attention with regard to improving the actuation performance, stability, and electrochemical properties of the IEAP actuator. For the study of the actuator interface, the basic structure of IEAP should be noted.

The structure of an IEAP actuator is similar to that of a capacitor, which stores energy in the form of electrical charge [18,19]. This has led to some researchers in the IEAP field to report the capacitance results of actuators [13]. For example, Akle et al. [20] reported a linear correlation between the strain response and capacitance of ionomeric materials. However, the capacitance and strain values of actuators require further improvement, and there have been few studies of the relationship between the driving performance and capacitance of IEAP actuators. Known methods of improving capacitance include the use of electrolyte polymers, insulators, and carbon materials [21,22,23]; we suggest that the electrochemical and driving properties of IEAP actuators could be improved by placing insulating layers between the polymer membrane and electrodes, forming a structure similar to that of a traditional capacitor. As the insulator, we suggest hexagonal boron nitride (h-BN). This chemically stable material, which has a two-dimensional structure, excellent thermal conductivity (~400 W m^−1^ K^−1^), and a wide band gap of ~6 eV [24,25,26,27,28,29], has been reported to block large ions under an applied voltage [30]; these characteristics indicate that it would be a suitable insulating material for use in IEAP actuators. 

IEAP actuators work because ion hydrates and water molecules move within the ionic polymer membrane upon application of an external voltage. Unfortunately, as the actuator is operated, the ion hydrates evaporate because of the heat generated by the electrodes; this reduces durability. However, water molecules are larger (>0.28 nm) than the lattice constant of h-BN (*a*_h-BN_ = 0.25 nm); therefore, their evaporation from the ionic polymer can be efficiently blocked by h-BN insulating layers.

In this study, we developed and tested an IEAP actuator with insulating layers of h-BN particles between the ionic polymer membrane and the top and bottom electrodes. The insulating properties of h-BN particles of different sizes were systematically tested to determine the optimal performance. We observed the changes in the driving and electrochemical properties according to the h-BN particle size and compared the results with those obtained using a conventional IEAP actuator without insulating layers. The IEAP actuator with h-BN insulating layers (particle size = 170 nm) exhibited improved capacitance and ion conductivity compared to the actuator without h-BN insulating layers; thus, the driving characteristics were improved. In addition, the layered actuator had high energy and power densities and exhibited stable gripping abilities even when subjected to external vibration and movement, suggesting that IEAP actuators with insulating layers could be utilized in soft robotics and the flexible actuator field as well as in energy storage.

## 2. Results

We designed an IEAP actuator (Figure 1) with an ionic polymer (Nafion-117) membrane, P/GO–Ag electrodes, and insulating layers between the polymer membrane and electrodes composed of h-BN powders with particle sizes of 1 μm, 800 nm, 300 nm, or 170 nm (denoted h-BN−1 μm, h-BN−800 nm, h-BN−300 nm, and h-BN−170 nm, respectively), in a Nafion matrix. The P/GO–Ag paper electrode and h-BN insulating layers were included to increase the mobility of ions in the Nafion membrane and decrease the interface resistance between the electrodes and Nafion membrane, respectively.

Appendix A shows the morphological properties of the P/GO–Ag paper electrode. The Ag NWs within the electrode were well connected, as shown in Appendix A. The average thickness of the P/GO–Ag electrode was ~13 μm (Appendix A), and the surface was smooth, with an average surface roughness of ~88 nm (Appendix A). The sheet resistance was uniform across the entire electrode surface, with a value of ~200 mΩ sq^−1^. The h-BN powder was well dispersed without aggregation within the insulating layer (Figure 2a–d), particularly for smaller h-BN particle sizes; the roughness of the insulating layer decreased with decreasing h-BN particle size (Appendix A). The P/GO–Ag electrode and Nafion-117 membrane became more smoothly attached as the h-BN particle size decreased (Figure 2e–h). These results suggest that the h-BN−170-nm insulating layer, which was thin and uniform on both sides of the Nafion membrane, may have reduced ion intercalation to the PEDOT:PSS layers of the electrode. This would increase the capacitance and water uptake (WUP) of the actuator by blocking the leakage of ions from the Nafion membrane through the P/GO–Ag electrodes.

We also investigated the effect of these layers on the ion capacitance and driving performance. An IEAP actuator with insulating layers between the electrodes and ionic polymer membrane can be regarded as a capacitor. We measured the electrochemical properties of the IEAP actuators with and without h-BN insulating layers and observed the effect of the h-BN insulating layers on the current–potential curves and capacitance. Cyclic voltammetry (CV) was conducted in the applied voltage range of −1.0 to +1.0 V at a scan rate of 50 mV s^−1^, revealing quasi-ideal parallel-plate capacitive behavior.

The CV curves differed with the size of the h-BN particles. The actuators without h-BN layers produced rectangular CV curves (Figure 3a and Appendix A). No redox peak was observed and the specific capacitance was 0.5 mF g^−1^ at a scan rate of 50 mV s^−1^. By contrast, the actuators with h-BN layers produced redox peaks in both the negative and positive regions (−0.4 to +0.4 V) because of the redox reaction at the h-BN surface. The area of the CV curve was much larger for the IEAP actuators with h-BN insulating layers than for those without them; it increased as the h-BN particle size decreased.

The total area of the CV curve corresponds to the capacitance. The average capacitances are presented in Appendix A, and the average specific capacitances considering the weight of the IEAP actuators are presented in Figure 3b. The IEAP actuators with h-BN insulating layers had much higher specific capacitances than those without, and the specific capacitance increased with decreasing h-BN particle size. In particular, the IEAP actuator with h-BN−170 nm insulating layers exhibited greater capacitance (4.020 × 10^−1^ F) and specific capacitance (5.83 F g^−1^) than the IEAP actuator without insulating layers. These results are sufficient to show that the IEAP actuator has excellent capacity performance and high energy efficiency within the Nafion membrane [31,32,33].

The electrochemical impedance of the fabricated IEAP actuators was measured in the frequency range from 1 Hz to 100 kHz. Figure 3c shows Nyquist plots of the IEAP actuators. Each Nyquist plot comprises a semicircle and line. The presence of the semicircle demonstrates that the system has an equivalent electrical circuit (inset in Figure 3c) with electrolyte resistance *R*_e_ and double-layer capacitance *C*_dl_ in parallel at high frequencies [34,35]. The ion conductivity (*σ*_i_) was calculated from the electrolyte resistance of the IEAP actuator (Equation (S2)); it increases as the electrolyte resistance of the actuator decreases. The electrolyte resistance represents the interface resistance between the electrode and ion polymer film, and a decrease in interface resistance can enhance the ion conductivity of the ion polymer film. The ion conductivities of the IEAP actuators with h-BN insulating layers increased with decreasing h-BN particle size (Figure 3b). The ion conductivity of the IEAP actuator with h-BN−170-nm insulating layers was approximately twice that of the IEAP actuator with none.

We measured the driving performances of the IEAP actuators according to the particle size of the h-BN powder. The driving performance was also evaluated as a function of the capacitance. In addition, the weights of the IEAP actuators before and after the ion-substitution process were measured to assess the WUP within the Nafion membrane (Equation (S3)), thus obtaining information on the amount of water molecules and ion hydrates contained in the Nafion membrane. The weights and WUPs of the IEAP actuators are presented in Table 1. The IEAP actuator with h-BN−170-nm insulating layers had the highest WUP of the fabricated actuators (32.69%), because the uniform formation of the insulating layers across the electrode/Nafion membrane interface reduced the ion intercalation of the PEDOT:PSS layer and ensured the ions were retained inside the Nafion membrane. Table 1 also presents the weights of the IEAP actuators before and after actuation. The weight loss of the IEAP actuator with h-BN−170-nm insulating layers was ~2.9%, which is one-ninth that of the IEAP actuator without insulating layers (27.54%). The results suggest that the 170 nm h-BN particles form a uniform interface layer, improving the durability of the IEAP actuator because they are smaller than water molecules and therefore inhibit evaporation from the Nafion membrane during operation.

The Nafion membrane of the IEAP actuator with h-BN−170-nm insulating layers contained 50% more ions than that of the actuator without insulating layers (see WUP in Table 1). Therefore, the insulating layers may improve the driving properties. The driving performances of the IEAP actuators were measured under a sinusoidal input with a peak voltage of 2.5 V and excitation frequency of 0.2 Hz (Figure 4a,b). The IEAP actuator with h-BN−170 nm insulating layers exhibited a larger displacement (3.21 mm) and peak-to-peak value (6.01 mm) than the other IEAP actuators. In agreement with the weight-loss results, the IEAP actuator with h-BN−170 nm insulating layers had better durability than the other IEAP actuators: it showed only slight degradation (within 4.5%) of the peak-to-peak value over 5 h, compared to the drastic 96.3% degradation over 5 h of the IEAP actuator without insulating layers (Figure 4b). Its curvature performance (35.59 m^−1^) was also better than those of the other IEAP actuators and ~7.4 times greater than that of the actuator without insulating layers. Figure 4d shows the actual movement of the IEAP actuators after 60 s under 2.5 V_DC_ input. These results show that the h-BN insulating layers reduced the interface resistance, thus increasing the ion storage capacitance and ion conductivity in the Nafion-polymer membrane. The driving characteristics improved accordingly. Appendix A shows the driving performance of the IEAP actuator with h-BN−170-nm insulating layers under various voltages and frequencies.

## 3. Discussion

The surface resistance, capacitance, and driving characteristics of the actuators obtained in this study are compared with reported results [36,37,38,39,40,41] in Table 2. The maximum strain value of the IEAP actuator with h-BN−170-nm insulating layers was 8.31%, larger than those of the earlier actuators (the calculation details [42] are provided in the Appendix A). In addition, the electrode surface resistance and capacitance of the developed actuator were superior to those of earlier actuators. Most previous studies did not investigate both the capacitance and driving characteristics of the actuator, and none mention the correlation between them reported here. Figure 5 shows that the displacement and curvature of the IEAP actuators increase with increasing capacitance. Moreover, the capacitance and driving performance of IEAP actuators increase with decreasing h-BN particle size. In particular, the capacitance of the IEAP actuator with h-BN−170-nm insulating layers is more than 10^5^ times that of the IEAP actuator without insulating layers. These results highlight that the h-BN insulating layers increase the capacitance of the IEAP actuator, with a positive effect on the driving performance.

As shown in Figure 6a, different potential application sectors of IEAP actuators have distinct power-density and energy-density requirements; actuators without insulating layers do not satisfy any of them. We investigated the power densities (P) and energy densities (E) of the IEAP actuators with h-BN insulating layers (Figure 6a). The IEAP actuators with h-BN insulating layers had supercapacitor characteristics [41]. As the h-BN particle size decreased from 1 μm to 170 nm, P increased from 23.97 to 606.88 W kg^−1^, whereas E increased from 0.2 to 5.06 Wh kg^−1^. In particular, the P and E values of the IEAP actuator with h-BN−170 nm insulating layers were superior (P: 606.88 W kg^−1^, E: 5.06 Wh kg^−1^) to those of the IEAP actuator without insulating layers (P: 0.054 W kg^−1^, E: 4.54 × 10−4 Wh kg^−1^).

A simple device was developed to illustrate the applicability of the IEAP actuator with h-BN−170 nm insulating layers. A two-finger gripper composed of two IEAP cantilevers (IEAP-actuator sheets) was designed to imitate the grasping motion of fingers; its operation is shown in Figure 6d, e, as well as Appendix A. Figure 6b shows the two metal plates used to create the gripper. The two-finger gripper had a length and width of 23 and 10 mm, respectively, and a weight of 0.087 g (Figure 6c). Given the importance of water molecules and cations inside the polymer membrane for actuator operation, working voltage conditions that would not cause water decomposition (i.e., <1.23 V) were required; therefore, the actuator was operated at 1 V. The performance of the two-finger gripper on a 1.5 g plastic ball is shown in Figure 6d. In fact, the two-finger gripper was able to maintain a stable hold on objects that weighed more than 8.6 times as much as the actuator device, despite strong external vibration and fast or slow movement (Appendix A).

The high capacitance, high driving performance, and minimal leakage of vaporized water molecules and cations during electrical stimuli exhibited by IEAP actuators with h-BN−170 nm insulating layers suggest other possible applications, including biomedical devices [42,45] and biomimetic soft robots [46,47]. Moreover, the power and energy density results show that these actuators could be used as supercapacitors [43].

## 4. Materials and Methods

### 4.1. Materials and Characterization

We used deionized (DI) water (resistivity: >18 MΩ cm at 25 °C) in all experiments. Poly(3,4-ethylenedioxythiophene):poly(styrenesulfonate) (PEDOT:PSS) (1.3 wt%) and Triton X-100 were purchased from Sigma-Aldrich (St. Louis, MO, USA). Graphene oxide (GO) was synthesized by the Hummers and Hoffman methods, and Ag nanowires (NWs) were purchased from DUKSAN Hi-Metal (Ulsan, Korea). Cellulose-membrane filter papers (pore size: 0.20 µm, diameter: 47 mm) were purchased from HYUNDAI MICRO., Ltd. (Anseong, Korea). The PEDOT:PSS, GO, Ag NWs, and cellulose filter papers were used to fabricate PEDOT:PSS/GO–Ag (P/GO–Ag) NW paper electrodes for the IEAP actuator. Sulfonated-tetrafluoroethylene-based fluoropolymer-copolymer membranes (thickness: ~183 µm; Nafion 117) and Nafion solution were purchased from Dupont (Wilmington, DE, USA). These were used as the polymer membranes of the IEAP actuator and the attachment resin between the polymer membranes and P/GO–Ag electrodes, respectively. Insulating layers for the IEAP actuators were formed from h-BN powders with different particle sizes (1 μm, 800 nm, 300 nm, and 170 nm), purchased from Ditto Technology (DT-BN-20PG, Korea). Lithium chloride (LiCl) purchased from Sigma-Aldrich (Seoul, Korea) and 1-ethyl-3-methylimidazolium trifluoromethanesulfanate (EMIM-OTf) purchased from Merck (Seoul, Korea) were used in the ion-substitution process to produce the actuator. We fabricated an h-BN/Nafion mixture using an ultrasonic probe sonicator (UW200, Bandelin, Germany) to ensure that the h-BN powder (0.5 wt%) was well dispersed in the Nafion solution. The morphological properties of the P/GO–Ag electrode, h-BN insulating layer, and IEAP actuator were investigated using field-emission scanning electron microscopy (S-4300, Hitachi, Japan) and atomic force microscopy (non-contact mode, XE-100, Parksystems, Suwon, Korea). The sheet resistance of the P/GO–Ag electrodes was measured using a four-point probe system (DASOL, ENG, FPP-HS 8). The electrochemical properties of the IEAP actuator were measured using cyclic voltammetry (CompactStat, HS Technology, Korea), and the current density, capacitance, and impedance were analyzed. The driving performance was measured using a laser displacement sensor (OMRON, ZS-LD80; beam length: 0.9 mm, beam thickness: 60 µm).

### 4.2. Fabrication of IEAP Actuator with Insulating Layers

First, we incorporated Li+ and EMIM-OTf cations into the Nafion-117 membrane by an ion exchange process using LiCl (LiCl: 16 g, DI water: 244 mL) and EMIM-OTf (EMIM-OTf: 55 g, methanol: 28 g) solutions. Second, we mixed GO and Ag NWs in a 1:2.5 volume ratio, and then fabricated the GO–Ag NW electrodes by filtration using the cellulose filter paper. The filtered GO–Ag NW electrode was put into acetone to peel off the filter paper and dried in a vacuum oven at 100 °C. Subsequently, we coated the Nafion-117 membranes with the h-BN/Nafion mixture to create a uniform insulating layer. The GO–Ag NW electrodes were attached to the Nafion-117 membrane by hot-pressing at 0.1 MPa and 100 °C for 5 min. Finally, a mixture of Triton X-100 (0.766 mL) and PEDOT:PSS (10 mL) [27] was spin-coated onto both surfaces of the IEAP actuator. The finished P/GO–Ag electrode-based IEAP actuators with h-BN insulating layers were rectangular with dimensions 0.5 × 3.5 cm. An actuator sample without h-BN insulating layers was fabricated in a similar manner but without the h-BN/Nafion mixture coating.

## 5. Conclusions

In this study, IEAP actuators with h-BN insulating layers between the ionic polymer membrane and the top and bottom electrodes were fabricated and tested. We demonstrated for the first time that the driving performance of an actuator improves as its capacitance increases, highlighting the strong relationship between the capacitance and actuator performance. The IEAP actuator with h-BN−170-nm insulating layers showed the highest capacitance (4.020 × 10^−1^ F) and largest bending value (6.01 mm) of the tested IEAP actuators. The durability of this IEAP actuator (displacement degradation within 4.5%) was also greatly improved (by a factor of ~21.4) over that of the IEAP actuator without insulating layers (displacement degradation of 96.3%). We attribute the improved durability to the lattice constant of the h-BN particles being smaller than the size of the hydrated ions; this suppresses evaporation from the Nafion membrane during operation. Our results suggest a new direction for IEAP actuator research and may to contribute to the commercialization of actuators in the fields of intelligent robotics and biomimetic medical devices.

## Figures and Tables

**Figure 1 ijms-23-04981-f001:**
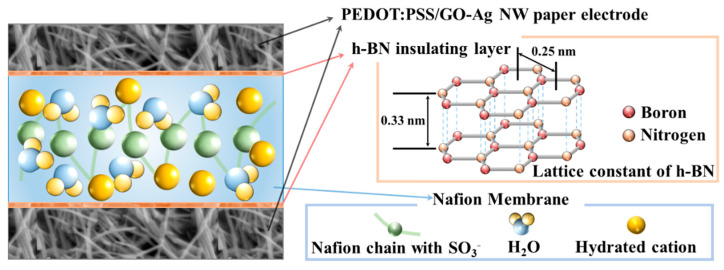
Schematic of ionic electroactive polymer (IEAP) actuator structure with hexagonal boron nitride (h-BN) insulating layers.

**Figure 2 ijms-23-04981-f002:**
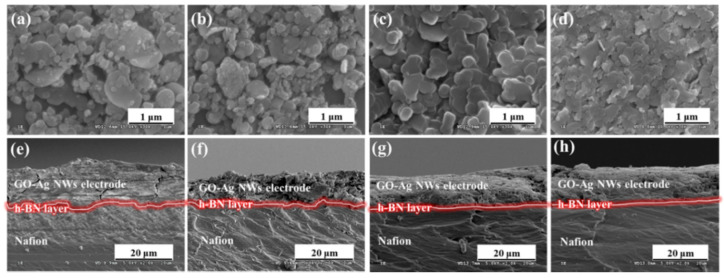
Top and cross-sectional scanning electron microscopy (SEM) images of IEAP actuators with h-BN insulating layers with different particle sizes: (**a**,**b**) 1 μm, (**c**,**d**) 800 nm, (**e**,**f**) 300 nm, and (**g**,**h**) 170 nm.

**Figure 3 ijms-23-04981-f003:**
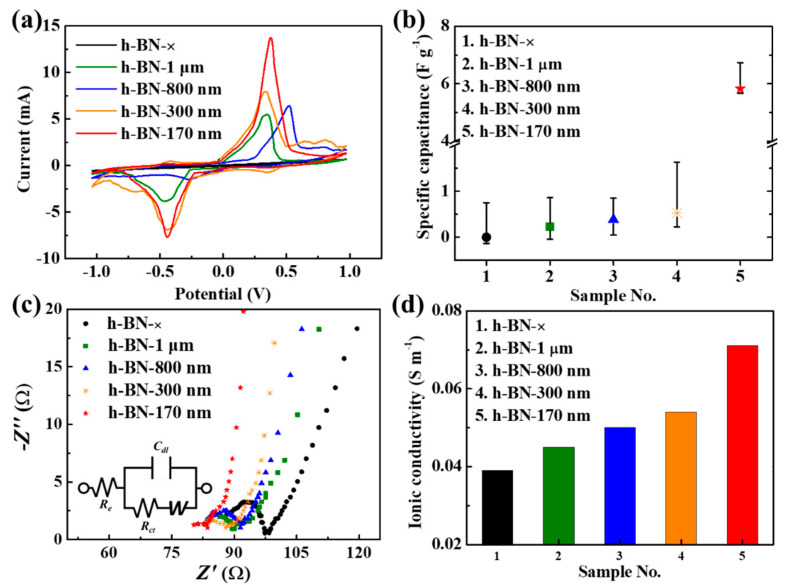
Electrochemical properties of IEAP actuators without insulating layers (h-BN−×) and with h-BN insulating layers with different particle sizes (1 μm, 800 nm, 300 nm, and 170 nm): (**a**) Cyclic voltammetry (CV) curves; (**b**) Specific capacitance; (**c**) Nyquist plots (inset: diagram of equivalent electrical circuit; *W* represents the Warburg diffusion element, *R*_e_ is the electrolyte resistance, *R*_ct_ is the polarization resistance, and *C*_dl_ is the double-layer capacitance); (**d**) Ionic conductivity.

**Figure 4 ijms-23-04981-f004:**
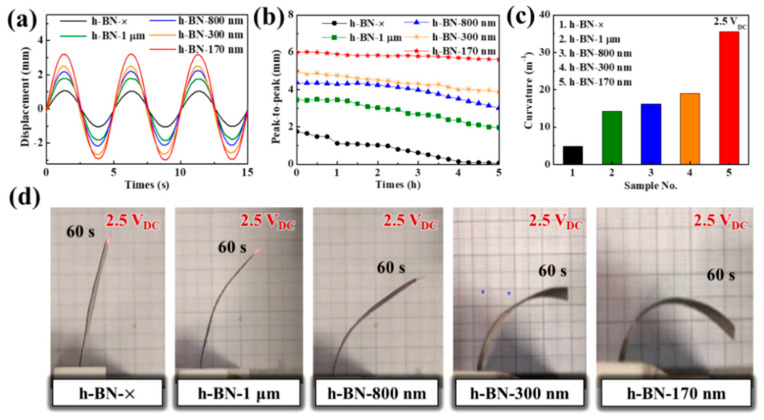
Performances of IEAP actuators without insulating layers (h-BN−×) and with h-BN insulating layers with different particle sizes (1 μm, 800 nm, 300 nm, and 170 nm): (**a**) Displacement at 0.2 Hz and 2.5 V_AC_; (**b**) Peak-to-peak value at 0.2 Hz and 2.5 V_AC_; (**c**) Curvature at 2.5 V_DC_; (**d**) Photographs of movement at 2.5 V_DC_.

**Figure 5 ijms-23-04981-f005:**
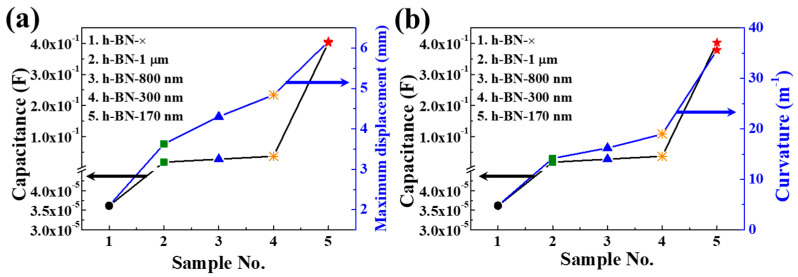
Effect of h-BN layer thickness on capacitance and driving performance of IEAP actuators without insulating layers (h-BN−×) and with h-BN insulating layers with different particle sizes (1 μm, 800 nm, 300 nm, and 170 nm): (**a**) Capacitance vs. maximum displacement; (**b**) Capacitance vs. curvature.

**Figure 6 ijms-23-04981-f006:**
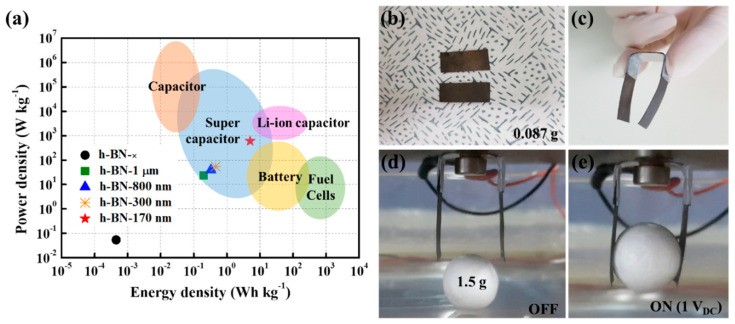
(**a**) Average power and energy densities of IEAP actuators without insulating layers (h-BN−×) and with h-BN insulating layers with different particle sizes (1 μm, 800 nm, 300 nm, and 170 nm) compared to requirements of various industrial fields. Images of (**b**) sheets of IEAP actuator with h-BN−170 nm used in (**c**) two-finger gripper. Images taken (**d**) before and (**e**) after grasping an object at 1.0 V_DC_.

**Table 1 ijms-23-04981-t001:** Water uptake (WUP) and weight loss (WL) of IEAP actuators without and with h-BN insulating layers.

Sample	*W*_d_^1^ [g]	*W*_s_^2^ [g]	*W*_f_^3^ [g]	WUP ^4^ [%]	WL ^5^ [%]
h-BN−× ^6^	0.056	0.069	0.050	23.21	27.54
h-BN−1 μm ^7^	0.059	0.074	0.063	25.42	14.86
h-BN−800 nm ^7^	0.056	0.071	0.065	26.79	8.45
h-BN−300 nm ^7^	0.054	0.069	0.065	27.78	5.80
h-BN−170 nm ^7^	0.052	0.069	0.067	32.69	2.90

^1^ *W*_d_: Initial (dry) weight of IEAP actuator. ^2^ *W*_s_: Weight of IEAP actuator after ion substitution of water molecules and ion hydrates in ionic polymer. ^3^ *W*_f_: Weight of IEAP actuator after actuation. ^4^ WUP: Water uptake during ion substitution {WUP = [(*W*_s_ − *W*_d_)/*W*_d_] × 100}. ^5^ WL: Weight loss during actuation {WL = [(*W*_s_ − *W*_f_)/*W*_s_] × 100}. ^6^ IEAP actuator without insulating layers. ^7^ IEAP actuators with insulating layers comprising h-BN powder with a particle size of 1 μm, 800 nm, 300 nm, or 170 nm in a Nafion matrix.

**Table 2 ijms-23-04981-t002:** Electrode, capacitance, and driving characteristics of various actuators from prior reports and the IEAP actuator with h-BN−170-nm insulating layers.

Actuator Types	Electrode Sheet Resistance [Ω sq.^−1^]	Specific Capacitance [F g^−1^]	Max. Displacement [mm] (at 2.5 VAC)	Curvature [m^−1^]	Strain [%]	Ref.
P/GO-Ag IEAP ^1^ Actuator with h-BN−170-nm Insulating Layers	2.00 × 10^−5^	5.83	6.01	35.59	8.31	This work
Au	<100	10^−4^	-	-	1.04	[5]
Pt	10	7.5 × 10^−4^ [F cm^−2^]	-	-	5	[10]
Ag nanopowder	0.12–0.15	-	5.00	-	-	[14]
Graphene	-	0.95 [meq/g]	0.145 (at 0.5 VAC)	-	-	[15]
Ionomer	-	-	5.00	-	-	[16]
IPMC	-	-	-	0.2 × 10^−3^	-	[17]
Ionomeric-IL	-	(1–5) × 10^−3^ [F cm^−2^]	-	-	2.44	[43]
PSS-*b*-PMB	-	0.12	4.00 (at 3.0 VAC)	-	4	[44]

^1^ PEDOS:PSS/graphene oxide–Ag nanowire electrode-based ionic electroactive polymer.

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
