# Peer review of "Effects of Hexagonal Boron Nitride Insulating Layers on the Driving Performance of Ionic Electroactive Polymer Actuators for Light-Weight Artificial Muscles"

_ijms, 2022, doi:10.3390/ijms23094981_

Round 1

Reviewer 1 Report

The article entitled Effects of hexagonal boron nitride insulating layers on the driving performance of ionic electroactive polymer actuators for light-weight artificial muscles is in my opinion prepared very well and can be published in IJMS. But I have two recommendations for the authors. First: The first paragraph in the introduction is too strict. It would be good to write it down more. Secondly, I recommend including Table S1 directly in the main article (text). The first thing I expected was a comparison with other results that had already been achieved.

Author Response

Thank you for your comment about our paper. 

Reviewer 2 Report

The authors presented a detailed study regarding the significant impact of the microstructure of the composite insulator layer in electrochemical actuators. The authors demonstrated that, this composite layer if designed effectively can block unwanted ion diffusion, provide thermal stability, and significantly enhance electrochemical energy stored in these actuator systems. This is a conclusive work that clearly shows constructing these composites from smaller hexagonal boron nitride (h-BN) fillers leads to much better properties and performance in the resulting actuators. However, there are some parts that needs clarification that are provided below:

1- Did authors tested a composite insulator system that consists of h-BN fillers smaller than 170 nm or is this a limitation for the synthesis method? This needs to be addressed since the results suggests an improving performance with decreasing filler size, which stops at 170 nm. 

2-Another important missing information is the thickness of the insulator layer of the actuators. It would be beneficial to include any study or comment on the actual thickness of the composite insulator layer employed in these actuators. Is this thickness optimized through experiments or theoretical estimations on ion diffusion?

3- This study can definitely benefit from a thermogravimetric analysis (TGA) study performed on the insulator layer, which not only reveals the composition of the composite insulator layer but also clearly presents the thermal stability of these composite materials.

4- Finally, authors provided a detailed an elaborate picture on the electrochemical energy stored in these actuator systems. But, a plot that presents actuation efficiency of these electrochemical actuators, which consists of curvature as a function of input power is necessary (having data representing each filler (h-BN) size in these actuator systems) . This will present what portion of the electrochemical energy is translated into mechanical work. Beyond analytical purposes, this will also provide a merit to compare these electrochemical actuators with respect to thermal (bimorph) actuators, which also commonly employ 2D or platelet filler materials. Please refer to following articles (similarly employing composite materials for thermal management) and more as reference for comparative analysis between electrochemical and bimorph actuators regarding actuation efficiency:

a) Amjadi, M.Sitti, M.Adv. Sci. 20185, 1800239.

b) M. Vural, Y. Lei, A. Pena-Francesch, H. Jung, B. Allen, M. Terrones,
M. C. Demirel, Carbon N.Y. 2017, 118, 404.

This will surely help improve the clarity and broaden the audience of this nice work.

Author Response

(The authors gave the same response as above.)
